# Aerobic Exercise and Stretching as Add-On to Inpatient Treatment for Depression Have No Differential Effects on Stress-Axis Activity, Serum-BDNF, TNF-Alpha and Objective Sleep Measures

**DOI:** 10.3390/brainsci11040411

**Published:** 2021-03-24

**Authors:** Christian Imboden, Markus Gerber, Johannes Beck, Anne Eckert, Imane Lejri, Uwe Pühse, Edith Holsboer-Trachsler, Martin Hatzinger

**Affiliations:** 1Psychiatric Services Solothurn, 4503 Solothurn, Switzerland and University of Basel, 4031 Basel, Switzerland; Martin.Hatzinger@spital.so.ch; 2Private Clinic Wyss, 3053 Muenchenbuchsee, Switzerland; 3Department of Sport, Exercise and Health, University of Basel, 4052 Basel, Switzerland; markus.gerber@unibas.ch (M.G.); uwe.puehse@unibas.ch (U.P.); 4Psychiatric University Hospital, University of Basel, 4031 Basel, Switzerland; Johannes.Beck@sonnenhalde.ch (J.B.); anne.eckert@upk.ch (A.E.); imane.lejri@upk.ch (I.L.); edith.holsboer@gmail.com (E.H.-T.); 5Private Clinic Sonnenhalde, 4125 Riehen, Switzerland

**Keywords:** aerobic exercise, depression, randomized controlled trial, HPA-Axis, BDNF, sleep, cardiorespiratory fitness

## Abstract

(1) Background: While the antidepressant effects of aerobic exercise (AE) are well documented, fewer studies have examined impact of AE as an add-on treatment. Moreover, various effects on neurobiological variables have been suggested. This study examines effects of AE on Cortisol Awakening Reaction (CAR), serum Brain Derived Neurotrophic Factor (sBDNF), Tumor Necrosis Factor alpha (TNF-alpha) and sleep. (2) Methods: Inpatients with moderate-to-severe depression (*N* = 43) were randomly assigned to the AE or stretching condition (active control) taking place 3x/week for 6 weeks. CAR, sBDNF and TNF-alpha were assessed at baseline, after 2 weeks and post-intervention. The 17-item Hamilton Depression Rating Scale (HDRS17), subjective sleep quality measured by the Pittsburgh Sleep Quality Index (PSQI) and polysomnography (PSG) were obtained at baseline and post-intervention. (3) Results: Stress axis activity decreased in both groups from baseline to post-intervention. sBDNF showed a significant increase over time, whereas the number of awakenings significantly decreased. No significant time by group interactions were detected for any of the study variables. Correlational analyses showed that higher improvements in maximum oxygen capacity (VO_2_max) from baseline to post-intervention were associated with reduced scores on the HDRS17, PSQI and REM-latency post-intervention. (4) Conclusions: While some neurobiological variables improved during inpatient treatment (CAR, sBDNF), no evidence was found for differential effects between AE and an active control condition (stretching). However, patients in which cardiorespiratory fitness increased showed higher improvements in depression severity and depression-related sleep-parameters.

## 1. Introduction

Depression is one of the most important causes of disability worldwide with still growing numbers within the last 20 years [1]. Due to increased all-cause morbidity and mortality, depression leads to a significant reduction of life expectancy [2] and therefore has a large impact on public health. Research efforts aimed at understanding its biological underpinnings and identifying effective ways to manipulate it has high priority.

Depression is associated with a variety of neurobiological features that might serve as biological markers. These include (a) a dysregulation of the hypothalamic-pituitary-adrenal axis (HPA-axis) with over secretion of stress hormones and reduced negative feedback [3]; (b) impaired sleep with reduced sleep continuity and increased rapid eye movement (REM) pressure [4]; (c) reduced expression of serum Brain Derived Neurotrophic Factor (sBDNF) [5] and (d) changes in inflammatory cytokines such as Tumor Necrosis Factor alpha (TNF- alpha) [6].

In various meta-analyses different exercise treatments especially aerobic exercise (AE) proved to have antidepressant effects improving symptom severity in mild to moderate depression [7] during various treatment settings [8]. AE might even improve cognitive symptoms of depression such as working memory [9,10,11], hinting at possible neurotrophic effects. Neurotrophic effects of exercise could be partly mediated by sBDNF [12,13]. In healthy controls, regular AE has shown to increase resting sBDNF [14]. However, evidence regarding the effects of AE on sBDNF in depression is still inconclusive. Two studies failed to find a significant effect of regular AE on sBDNF [15,16] while another trial with a small sample of primarily inactive depressed patients found a significant increase of sBDNF in the exercise group that was associated with improved depression scores and sleep quality [17]. Preclinical models have shown negative effects of stress on sBDNF, on the other hand, several antidepressants increase BDNF in the brain [18]. Stress has also been linked to decreased sBDNF and increased insomnia severity in humans [19], whereas sBDNF seems to play the role of a moderator between stress and insomnia. This all adds up to a possible link between stress, depression, sBDNF and insomnia.

Single exercise bouts have shown to activate HPA-axis: Vigorous exercise leads to increased cortisol secretion in healthy young adults reflected by increased hair cortisol content [20]. In depressed patients, cortisol response to an incremental bicycle test has shown to lead to a significantly lower cortisol response than in healthy controls [21]. Contrasting those findings, Kiive et al. [22] observed equal cortisol responses in depressed patients and healthy controls after an incremental bicycle test. Ida et al. [23] examined a sample of depressed patients in remission or with only mild depressive symptoms and found saliva free cortisol levels and depressive symptoms to decrease after a short 15 min exercise session. Regular AE might have the potential to reduce cortisol levels in depressed patients as could be shown by Foley et al. [24] who found a decrease of cortisol awakening response (CAR) in depressed patients after 12 weeks of AE. In another study with depressed patients, 12 weeks of AE resulted in a marked flattening of the diurnal cortisol slope and reduced cortisol levels at awakening [25].

Increased inflammation and elevated cytokines have shown to have several negative neurobiological effects including oxidative stress and dysregulation of HPA-axis. A bidirectional relationship between inflammation and depression has been proposed [26]. Meta-analyses have shown elevation of several cytokines such as TNF-alpha in patients with depression compared to healthy controls [27]. There also seems to be a strong link between increased inflammation and impaired neurogenesis associated with decreased BDNF expression in the brain, adding to the association between inflammation, neurogenesis and depression [28]. On the other hand, patients with chronic inflammation such as in the context of Crohn’s disease [29] and rheumatoid arthritis [30] show significantly increased levels of sBDNF. Since patients with chronic inflammatory disease are at increased risk for depression [31], this hints at a more complex interaction between inflammation, neurotrophins and depressive disorders. In the TREAD-study [32], two different exercise regimes were evaluated in a population of depressed patients failing to remit after treatment with selective serotonine-reuptake-inhibitors (SSRI). The authors found no changes of cytokines over the intervention period, but higher TNF-alpha levels at baseline were associated with larger decrease in depressive symptoms.

Additionally, exercise training has shown to have a wide range of positive effects on sleep. According to a large meta-analysis on the effects of physical activity on sleep in the general population, regular exercise participation had small beneficial effects on total sleep time (TST) and sleep efficiency and larger effects on sleep onset latency (SOL) and sleep quality [33]. Lederman et al. [34] conducted a meta-analysis on the effects of regular exercise on sleep quality in individuals with mental illness. Across *n* = 8 randomized-controlled-trials (RCTs) they found a large effect on sleep quality. However, the population of included studies suffered from various diagnoses such as substance use disorders, general anxiety disorder, depression and posttraumatic stress disorder and none of the included studies examined objective sleep measures such as polysomnography (PSG) or actigraphy. A recent network meta-analysis on the effects of exercise on sleep quality in unipolar depression reported significant improvement of sleep quality for several exercise interventions but not moderate AE alone if compared to passive control conditions. Compared to treatment as usual only mind-body exercise and vigorous strength exercise showed additional effects on sleep quality [35].

The purpose of our study was to examine the effects of regular AE on the abovementioned neurobiological systems affected by depression to better understand the antidepressant effect of AE. We hypothesized that regular AE in a moderately-to-severely depressed population of inpatients would (1) enhance neurogenesis and therefore increase sBDNF, (2) reduce HPA-axis dysregulation resulting in a decreased CAR, (3) decrease inflammation measured by TNF-alpha and (4) lead to improved subjective and objective sleep quality. In accordance with previously published data [36], we also aimed at analyzing (5) correlations between improved cardiorespiratory fitness and health-related outcomes such as depression severity and sleep-quality and (6) correlations between TNF-alpha levels at baseline and positive antidepressant effects of AE.

## 2. Materials and Methods

### 2.1. Participants and Procedures

The study was conducted as a two-armed RCT conducted at two sites, comparing AE to an active control intervention. Participants were recruited from the depression wards of two psychiatric hospitals in the German-speaking part of Switzerland. For randomization, lots were drawn for AE and the control condition in a 1:1 ratio by sex to achieve even distribution between males and females. The external exercise coaches who supervised all training sessions were informed about group allocation directly. Staff at the depression wards including all data assessors were blinded to group allocation. Blinding of patients was achieved by not telling them which intervention was considered as active treatment.

Screening took place during the first week of treatment and was conducted by the senior consultant of the respective ward. Inclusion criteria were: (a) aged ≥18 and ≤60 years; (b) being an inpatient on respective depression wards; (c) International classification of diseases 10 (ICD-10) diagnosis of depression (first episode, recurrent or bipolar; F32, F33, F31) with (d) a score >16 on the 17-item Hamilton depression rating scale (HDRS17); and (e) written informed consent. The following exclusion-criteria were applied: (a) any somatic condition not permitting regular AE; (b) body mass index (BMI) >35 kg/m^2^; (c) pregnant at baseline; (d) acute suicidal ideation; (e) comorbid substance addiction (except nicotine); (f) comorbid other major psychiatric disorder (such as schizophrenia); and (g) high-intensity exercise activities prior to admission (i.e., marathon-runners). Patients were recruited into the study between October 2013 and January 2016. To detect a moderate effect (*f* = 0.25) in the RCT on the primary outcome (depressive symptom-severity), a power analysis for repeated measures analyses of variance (ANOVAs) (using G*Power 3.1; alpha error probability: 0.05, power: 0.90, correlations among measures: 0.50) revealed that at least *n* = 36 (overall) participants are needed. We therefore originally aimed to include 40 patients with equal gender distribution [37].

Written informed consent was obtained from all subjects in accordance with the Declaration of Helsinki. Ethical approval was obtained by the respective local Ethics Committees: (a) Ethics Committee of both Basel (EKBB) in Basel, Switzerland (reference No. 62/13; obtained on 6 May 2013) and (b) Ethics Committee Aargau/Solothurn in Aarau, Switzerland (reference no. 2013/029, obtained 21 May 2013). Today the newly formed Ethics Committee of Northwestern and Central Switzerland (EKNZ) represents both former regional committees. We decided to assess TNF-alpha within the obtained biological samples after completing the study. Therefore, a protocol-amendment by the EKNZ (reference no. PB_2016-02488, obtained 14 November 2016) was obtained. The study was registered at www.clinicaltrials.gov (accessed on 18 March 2021) (trial number: NCT02679053). As we pointed out in the study protocol [37], the primary outcome of this study was depression severity. While the results of the primary outcome are already published elsewhere [11], the present paper focuses on the analysis of secondary outcomes (CAR, sBDNF, TNF-alpha and sleep data).

### 2.2. Intervention vs. Active Control Condition

The exercise intervention consisted of supervised AE on indoor bicycles three times per week. It lastet six weeks (reflecting median duration of inpatient treatment). The target heartrate (HR) was set at 60–75% of maximal heartrate (HRmax) monitored with Polar^TM^ RS800CX (Polar, Kempele, Finland). To estimate HRmax age-formula (220–age [in years]) was used. In accordance with Dunn et al. [38], the targeted weekly energy expenditure was defined as 17.5 kcal per kg bodyweight.

The active control condition was designed as a structured and standardized program three times per week for all major muscle groups. To ensure that the intensity in the control group was not too high, supervisors reminded participants not to get out of breath during trainings. This was important to distinguish between activity intensities of the intervention and active control group. To minimize the influence of social contact, all interventions were carried out with no more than two patients. A more detailed description of the interventions is published elsewhere [37].

All intervention sessions were conducted between 4 and 6 p.m. for approximately 40 to 50 min. Standard inpatient treatment for all patients consisted of pharmacological treatment according to Swiss national recommendations [39], individual and group-psychotherapy supported by a number of creative group therapies. However, participants were asked to refrain from any additional vigorous physical activities during the intervention period. Pharmacological treatment was limited to antidepressant treatment with SSRI or selective serotonine-norepinephrine-reuptake-inhibitors (SNRI) with lithium as a possible augmentation strategy. Antidepressant combinations, tricyclic antidepressants, monoamino oxidase inhibitors and antipsychotics were not allowed during the study period. An exception was quetiapine for sedation and augmentation treatment. Groups did not differ concerning pharmacotherapy [11].

### 2.3. Outcome Measures

At baseline, we assessed basic demographic data, smoking status and depression-specific variables (duration of current episode, number of previous episodes, age of onset). All patients underwent physical examination. An electrocardiogram (ECG) was obtained. Furthermore, bodyweight and height to calculate BMI, resting HR and blood pressure (BP) were measured. Symptom severity of depression was measured by a trained staff member with the HDRS17 [40].

Sleep data were assessed at baseline and after 6 weeks (post-intervention); details are provided below. All laboratory parameters (CAR, sBDNF and TNF-alpha) were measured at baseline, 2 weeks after the onset of the intervention period (+2 weeks) to detect early changes and at post-intervention. To monitor improvements in cardiorespiratory fitness and to estimate participant’s maximum oxygen capacity (VO_2_max), a Queens Step Test [41,42] was performed by all participants at baseline and at the end of every week.

#### 2.3.1. Cortisol Awakening Response (CAR)

CAR was measured at baseline, after 2 weeks and at post-intervention by obtaining salivary cortisol immediately after waking up, +10, +20 and +30 min later [43]. Participants were instructed to perform saliva-swaps at the defined times and not to eat, drink or smoke during the period of the CAR. Samples were obtained with commercially available devices (Salivette ^®^, Sarstedt, Germany). All samples were stored at −20 °C For analyses they were sent to the Biochemical Laboratory of the University of Trier (approximately every three months). Free salivary cortisol levels (nmol/L) were established, based on a time-resolved fluorescence immunoassay; for more details see: [44].

#### 2.3.2. Serum Brain Derived Neurotrophic Factor (sBDNF) and Tumor Necrosis Factor-Alpha (TNF-Alpha)

sBDNF and TNF-alpha were measured in serum samples (EDTA-blood) at baseline, after 2 weeks and at post-intervention. Due to diurnal variations in sBDNF [45], blood samples were obtained between 08 and 10 a.m. after overnight fasting according to a standardized protocol, using serum vacutainer tubes (Becton Dickinson). The serum tube was centrifuged at 1300× *g* for 10 min. Aliquoted samples were stored at −80 °C before assaying.

sBDNF levels were assessed using an enzyme-linked immunosorbent assay (ELISA) kit (Promega BDNF Emax^®^, lot 208463, Madison, WI, USA). Serum samples were appropriately diluted (1:100) and detection of BDNF was carried out in an antibody sandwich format as described in the manufacturer’s protocol (detection range: 7.8–500 pg/mL). The absorbance was measured within 30 min in a microplate reader set at 450 nm and a correction wave-length set to 690 nm, to determine BDNF concentrations according to the standard curve. All assays were carried out in duplicates and means were calculated.

Serum TNF-alpha levels were assessed using a solid phase enzyme amplified sensitivity immunoassay performed on microtiterplate (DRG^®^ TNF-alpha EIA-4146, lot 171402A, DRG International, Springfield, NJ, USA). 200 µL undiluted serum samples were assayed and detection of TNF-alpha was carried out in an antibody sandwich format as described in the manufacturer’s protocol (detection range: 0.7–446 pg/mL). The absorbance was measured in a microplate reader set at 450 nm against a reference wavelength set at 650 nm, to determine TNF-alpha concentrations according to the standard curve. In each assay, the results from the two internal quality controls were within the range specified on the respective vial label. All assays were carried out in duplicates and means were calculated.

#### 2.3.3. Sleep Data

Subjective sleep quality was assessed with the Pittsburgh Sleep Quality Index (PSQI) [46]. To measure objective sleep-parameters, we performed a PSG during one night using Compumedics somtéPSG with the canals electroencephalogram (EEG) C3 & C4, EOG, electromyogram (EMG) and ECG. PSG was performed at baseline and post-intervention (during the first week after the intervention has ended). To rule out any acute effects of exercise, PSG never was performed the night after a training session. Experienced sleep technicians performed visual sleep scoring according to the guidelines described by Rechtschaffen and Kales [47].

### 2.4. Statistical Analyses

To detect baseline differences between the intervention and control groups, ANOVAs were conducted and descriptive statistics were reported as M and SD. For categorical variables, we used Chi^2^ tests to examine group differences and reported the number (*n*) and frequencies (%) as descriptive statistics.

Changes in outcome variables over the time points were analyzed using repeated measures ANOVAs, with a between-subject factor group (AE vs. active control condition) and a within-subject factor time (baseline, 2 weeks, post-intervention; for sleep data: baseline, post-intervention). Statistical significance level was defined at an alpha level of 0.05.

As an index of cortisol reactivity, we calculated the area under the total response curve with respect to ground (AUC_Total_). This was achieved with the trapezoid formula described by Pruessner et al. [48], including the entire time period from waking up to 30 min later. We then calculated the total response curve with respect to increase (AUC_Net_) and the basal cortisol level over time (AUC_Basal_): AUC_Total_ = AUC_Net_ + AUC_Basal_.

To assess bivariate associations between outcome variables, correlational analyses with Pearson’s correlations (r) were conducted.

All analyses (except PSG data) were carried out with intention-to-treat (ITT). Primarily we aimed at multiple imputation to substitute missing values. However, probably due to the small sample size, this yielded highly improbable values. Therefore, we substituted missing values by last observation carried forward (LOCF). Because we could not obtain PSG data from all participants at baseline and because there was even less data available at post-intervention, we decided for a per protocol analysis for PSG data.

## 3. Results

### 3.1. Sample Description

Between October 2013 and January 2016, sixty-four patients were screened for eligibility, with 58 patients meeting the inclusion criteria. Of the eligible patients, 9 patients denied participation in the study due to the following reasons: no motivation for regular exercise (*n* = 6) and not wanting to undergo the necessary examinations (*n* = 3). Fourty-nine patients signed informed consent to participate in this study. Four patients withdrew consent prior to randomization (lacking motivation). Two further patients were excluded because they did not reach the required symptom severity (HDRS17 < 17) for inclusion at baseline. Forty-three patients were randomized to either AE (*n* = 22) or the active control condition (*n* = 21). One patient was discharged from the hospital before the assessment of baseline data. Therefore, a total of *n* = 42 patients was included in the ITT analyses (AE: *n* = 22, control: *n* = 20). Eight patients had to be excluded from the study during the intervention period (due to withdrawn consent [*n* = 2], early discharge [*n* = 2], attempted suicide [*n* = 1], or physical problems [*n* = 3]), resulting in a total of *n* = 34 patients who met all inclusion criteria and completed the intervention. Table 1 shows a detailed description of the sample. We found no statistically significant differences between the two intervention groups concerning HDRS17 scores, as well as diagnosis and smoking status. We also achieved an almost even sex-distribution (47.6% females in total sample). A more detailed description of the sample, including a participant flow chart has been published elsewhere [37].

### 3.2. Neurobiological Variables

#### 3.2.1. Cortisol Awakening Response (CAR)

Salivary cortisol during CAR increased significantly over time (from awakening to +30 min) at each time point (*p* < 0.001; baseline: Eta^2^ = 0.349, +2 weeks: Eta^2^ = 0.516, post-intervention: Eta^2^ = 0.541) without statistically significant between-group differences (baseline: *p* = 0.257, +2 weeks: *p* = 0.112, post-intervention: *p* = 0.637). Graphs are displayed in Figure 1. As shown in Table 2, AUC_total_ (*p* = 0.001; Eta^2^ = 0.307) and AUC_basal_ (*p* = 0.014; Eta^2^ = 0.201) each showed a large significant time effect regardless of group allocation, with the largest reduction during the first 2 weeks of the intervention period. For AUC_Basal_, we also found a time by group interaction of moderate effect size close to statistical significance (*p* = 0.094; Eta^2^ = 0.117). However, group by time interactions were not statistically significant.

#### 3.2.2. sBDNF and TNF-alpha

As shown in Table 2 and Figure 2, we found a significant increase over time for sBDNF (*p* = 0.014; Eta^2^ = 0.201), but not for TNF-alpha (*p* = 0.977). The main increase of BDNF took place during the first two weeks with an additional increase in the AE group until post-intervention. However, there was no significant time by group interaction.

### 3.3. Sleep Data

Most objectively measured data from PSG (sleep phases, sleep efficiency, TST and SOL) remained stable over the intervention period with neither time nor time by group effects. The only exception was number of awakenings where we found a significant reduction over time for both groups (*p* = 0.035; Eta^2^ = 0.166). Subjective sleep quality also showed a significant increase over time reflected by a large decrease of PSQI score (*p* < 0.001; Eta^2^ = 0.412), without evidence for a time by group interaction (for details see Table 3).

### 3.4. Correlations

Correlations are shown in Table 4. We found a significant positive correlation for TNF-alpha with PSQI at baseline (*p* = 0.039; *r* = 0.327). Change of cardiorespiratory fitness measured by VO_2_max (dVO_2_max as the difference of VO_2_max between baseline and post-intervention) showed significant moderate-to-large negative correlations with HDRS17 (*p* = 0.049; *r* = −0.322), PSQI (*p* = 0.024; *r* = −0.375) and REM latency (*p* = 0.010; *r* = −0.437) at post-intervention. Furthermore, we observed a significant correlation between PSQI-scores at baseline and changes in VO_2_max (*p* = 0.024; *r* = −0.375), indicating that VO_2_max increased less among patients who initially reported poor sleep.

## 4. Discussion

Our study showed that AE had no additional effects on CAR, sBDNF, TNF-alpha and sleep parameters. However, sBDNF and CAR improved significantly during inpatient treatment.

Our findings seemingly contrast with previous findings by Foley et al. [24] who found a significant reduction of CAR in depressed patients who were assigned to a 12-week AE program (*n* = 10) compared to stretching (*n* = 13). However, the authors stated that they had found a significant decrease of CAR for both groups (without group differences) at week 6. Therefore, our results are comparable and a longer intervention-period might have resulted in similar group effects. The authors did not provide detailed information on the exact exercise intensity and weekly amount of exercise, making it difficult to compare our results directly. Another study with a larger sample [25] also found reduced cortisol levels at awakening after 12 weeks of AE (*n* = 38) compared to treatment as usual (*n* = 27) and internet cognitive behavioral therapy (*n* = 56). The fact that their design lacks an active control intervention hinders a direct comparison with our results and it is plausible that the low intensity of our stretching program might already have had a comparable effect on CAR. Since we lack a third arm in our study with no intervention, this remains a hypothesis that needs further testing. Since we found an effect of moderate magnitude (close to statistical significance) for a time by group interaction for CAR_Basal_, another explanation might be that the power of our study was not high enough to detect differences between groups.

Our finding of an increase over time of sBDNF without significant group differences corroborates the findings of Schuch et al. [16] who reported results of a similar trial with depressed inpatients and a comparable exercise intervention. They also found a significant increase of sBDNF in both groups after 2 weeks, pointing towards a general effect of multimodal inpatient treatment on sBDNF. However, their intervention only lasted for 3–4 weeks and they lacked an active control. The study by Szuhany et al. [15] also reported no differences on BDNF-levels between an AE group and an active control group of *n* = 29 depressed (sedentary) patients. However, their main study outcome was an acute effect of exercise bouts on BDNF-levels. Taken together, the evidence about effects of AE on sBDNF in depressed patients is still inconclusive, which might be due to various confounding factors such as by antidepressant medication [49]. Another reason for the time effect we found in our sample might again be explained by the choice of an active comparator that might have been sufficient to lead to an increase of sBDNF by itself compared to no additional physical activity.

In our study, neither multimodal depression inpatient treatment nor AE by itself resulted in any effects on TNF-alpha. A meta-analysis on the effects of antidepressant medication on inflammatory markers supports this finding since it was also unable to identify a significant change in TNF-alpha by antidepressants [50]. Rethorst and coworkers [32] found a promising correlation between TNF-alpha levels at baseline and depressive symptoms after 12 weeks of AE, hinting at the possibility that patients with high TNF-alpha levels might benefit more from AE in depression treatment. Nevertheless, our study could not replicate their results despite a similar study design. Differences in the design consist of a shorter duration of the intervention (half as long), a smaller sample size and a different sample composition (inpatients in our study versus outpatients in their study). Therefore, future studies on moderating effects of TNF-alpha on the antidepressant effects of AE should aim at a larger sample and a longer intervention period. It also remains unclear if TNF-alpha is an adequate laboratory marker to detect anti-inflammatory effects of AE.

Our study found a large time effect for improved self-reported sleep quality measured by PSQI for both groups, but without any group differences. Since improving sleep-quality was an important aim in depression-treatment at both study centers, this probably reflects a general effect of the treatment setting. However, the meta-analysis of Lederman and colleagues [34] showed significant positive effects of exercise at various levels of intensity on (subjective) sleep quality in individuals with various mental diseases. Therefore, our active control intervention might already have had beneficial effects on sleep quality resulting in difficulties to detect a difference between groups. Since we did not include a third group without any exercise intervention (treatment as usual), this remains inconclusive. On the other hand, a recent network meta-analysis suggests that mind-body exercise and vigorous strength training are more efficacious in improving sleep quality in unipolar depression than AE [35]. These results strengthen the hypothesis that in our sample subjective sleep quality improved due to overall treatment effects.

In addition, a moderate time-effect on reduced number of nightly awakenings, PSG-data found no additional effects on sleep architecture, sleep-onset-latency or sleep efficiency. To our knowledge, this is the first trial to examine the effects of regular AE on PSG-data in depressed patients. Our results suggest no additional effects of AE on objective sleep measures, but due to the small sample size and the fact that PSG could not be obtained from all participants, our findings have to be interpreted with caution. In healthy populations, positive effects of regular exercise on objective sleep variables have been demonstrated in the meta-analysis of Kredlow et al. [33]: these authors showed small beneficial effects on TST and sleep efficiency and small-to-moderate effects on SOL, findings that we could not replicate in our depressed inpatient-sample.

Despite the fact that we reported similar antidepressant effects of AE and stretching exercise in a previous paper [11], our findings suggest that–regardless of exercise modality-improvement of VO_2_max is associated with reduced depressive symptoms. This finding is in line with the results reported by Gerber et al. [36] who also found a significant association of improvement of VO_2_max with fewer depressive symptoms in a sample randomized to either high-intensity interval-training or AE. Martinsen et al. [51] found similar results in one of the first studies on the effects of AE on depression reporting lower depressive symptoms after 9 weeks of AE in participants with higher increase in VO_2_max. Moreover, we also found correlations of improved fitness with better subjective sleep quality (as assessed with the PSQI) at baseline and post intervention. This suggests that good sleep might play a moderating role in the improvement of VO_2_max. Including parameters of sleep quality into further studies on the effects of exercise on depression might be of importance to further examine the role of sleep quality. Additionally, increased fitness was associated with increased REM-latency at post-intervention. To our knowledge, this is the first study to report such an association. Since reduced REM-latency is a distinct sleep-related parameter of depression [52], this finding might reflect at possible mechanisms linking effects of AE on fitness with possible beneficial effects on depression. On the other hand, REM-latency in our sample was increased at all timepoints compared to healthy controls who normally show REM-latency of approximately 60 min [53]. A probable explanation for the high REM-latency compared to healthy controls is the fact, that most antidepressants increase REM-latency by up to 200% [52]. Due to this confounding factor and the small sample size, this finding should be interpreted with caution. Future research is needed to shed further light onto the interplay between sleep, depression and exercise.

Our study has several limitations. The sample size of *n* = 42 in the ITT-sample is rather small to detect smaller effects on the studied neurobiological variables. This fact has even more weight since those variables were not our main endpoints. However, small sample sizes are a common problem in exercise trials for depression: a recent review and meta-analysis for exercise and depression in mental health services has reported a similar average sample size of *n* = 43 [8]. PSGs were only recorded once; therefore, we might have an increased chance to have first-night effects of sampling errors by not measuring regular sleep of participants. On the other hand, since we studied inpatients, the probability that sleep was impaired by an unusual environment is low. Additionally, there are various other factors influencing PSG results such as the equipment used to record it. It would, therefore, have been advisable to include sleep diaries to enhance the accuracy of our sleep data. Since we applied per-protocol analysis to the PSG-data due to the high number of missing PSGs, our study probably is substantially underpowered regarding objective sleep-data. The validity of the PSG-data might also be questionable since we obtained PSG from more patients at baseline than post-intervention. Since the PSQI evaluates sleep-quality across the last 4 weeks, we might not measure sleep exactly post-intervention, which somewhat limits the accuracy of the PSQI in our study. Another limitation is that patients had to obtain the saliva samples for CAR by themselves after verbal and written instructions and were not supervised in the process. This might lead to bias by (a) the possibility of not starting the assessment of the CAR immediately after awakening and (b) not following the exact time-schedule of the procedure [54]. We also deviate from two other recommendations of the CAR expert consensus guidelines by Stalder et al. [54]: since cortisol peaks might arise as late as 45 min after awakening, we might have missed the peak in some individuals. Therefore, our CAR values might be an underestimation of the “real” cortisol response after awakening. Additionally, it has been proposed to sample CAR during at least two consecutive days due to low test-retest reliability. Our study only includes CAR for one day per timepoint. We also acknowledge that we included *n* = 4 patients with bipolar depression into our study, which might have led to an additional risk of bias since bipolar depression might differ from unipolar depression neurobiologically. On the other hand, there is robust evidence of benefits from AE for patients with bipolar depression [55] and since conversion rates from uni- to bipolar depression can reach up to 8% per year [56], samples of unipolar depression usually are always heterogeneous. Since there are reports on neurotrophic effects of lithium [57], including patients with lithium-treatment also might add to heterogeneity concerning sBDNF. On the other hand, only few patients received lithium and the groups did not differ concerning pharmacotherapy as published elsewhere [11]. We further admit that the Queens College Step test provides only a relatively rough estimate of VO_2_max. While this method is easy to implement and has been validated previously [41], it is not the most accurate approach to get an exact estimation of VO_2_max. The exercise intensity in the active control group was defined by subjective parameters. Including objective parameters such as an actigraph would have allowed a more robust interpretation of the intensity level of the active control condition. Finally, our study design with an active comparator yields the risk of underestimating the effects of AE, since low-intensity activities (such as stretching) could trigger beneficial effects. Including a third study arm with “treatment as usual” would enhance our results and its interpretation. To our knowledge, this is the first study to incorporate PSG-data into a RCT on the effects of exercise treatment for depression.

## 5. Conclusions

While some neurobiological variables (CAR, sBDNF, awakenings) and subjective sleep quality improved during multimodal inpatient treatment of depression, no evidence was found for differential effects between AE and an active control condition (stretching). However, increasing patients’ cardiorespiratory fitness levels might have a positive effect on depression severity and depression-related sleep-parameters and might in turn be influenced by sleep quality.

## Figures and Tables

**Figure 1 brainsci-11-00411-f001:**
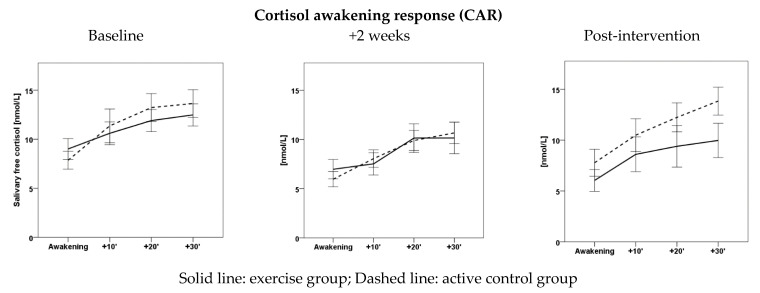
Cortisol awakening response for each timepoint by group. At each timepoint there was a significant increase of salivary cortisol (*p* < 0.001). Error bars represent standard error (SE).

**Figure 2 brainsci-11-00411-f002:**
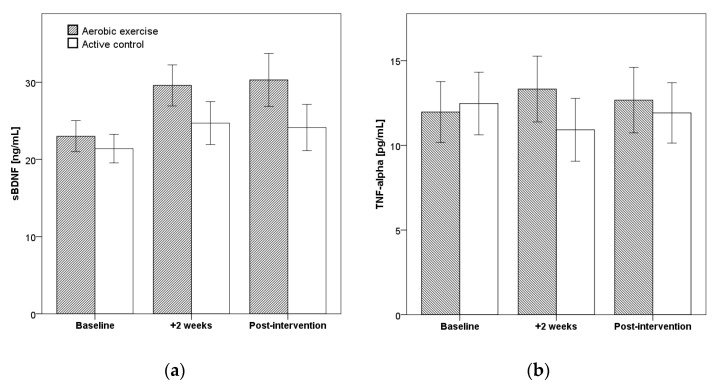
Serum levels of sBDNF (**a**) and TNF-alpha (**b**) at each timepoint by group. Error bars represent standard error (SE).

**Table 1 brainsci-11-00411-t001:** Sample description at baseline.

	Aerobic Exercise		Active Control Group		Total Sample		
*n*	22		20		42		
	*n*	%	*n*	%	*n*	%	p [Chi^2^ Test]
Females	10	45.5%	10	50.0%	20	47.6%	0.768
Smokers	7	31.8%	11	55.0%	18	42.9%	0.256
Diagnosis:							
First depression (F32)	10	45.5%	7	35.0%	17	40.5%	0.393
Recurrent depression (F33)	9	40.9%	12	60.0%	21	50.0%	
Bipolar depression (F31)	3	13.6%	1	5.0%	4	9.5%	
	***M***	***SD***	***M***	***SD***	***M***	***SD***	***p*** [ANOVA]
Age	41.3	(9.2)	38.3	(13.4)	39.9	(11.4)	0.404
Age at first episode	33.8	(12.1)	28.9	(13.6)	31.3	(13.0)	0.232
Duration of episode [weeks]	15.6	(15.2)	21.6	(23.6)	18.4	(19.6)	0.340
Prior depressive episodes [#]	1.9	(2.9)	3.6	(7.0)	2.7	(5.2)	0.465 ^(1)^
Prior maniac episodes [#]	0.3	(0.9)	0.0	(0.0)	0.1	(0.7)	0.165 ^(1)^
HDRS17	22.0	(4.0)	20.9	(2.6)	21.5	(3.4)	0.266
BMI [kg/m^2^]	25.9	(5.4)	23.9	(4.8)	25.0	(5.2)	0.227

Notes. Baseline values, *p*-values are based on ANOVAs and Chi^2^-tests. ^(1)^ ANOVA was conducted with logarithmized values due to severe violation of normality (skewness ≥ 2, kurtosis ≥ 7). HDRS17 = 17-item Hamilton Depression Rating Scale. BMI = Body mass index.

**Table 2 brainsci-11-00411-t002:** Neurobiological variables.

	Baseline				2 Weeks				Post Intervention									
	Aerobic Exercise		Active Control Group		Aerobic Exercise		Active Control Group		Aerobic Exercise		Active Control Group		Time			Time × Group		
	*M*	*SD*	*M*	*SD*	*M*	*SD*	*M*	*SD*	*M*	*SD*	*M*	*SD*	*F*	*p*	Eta^2^	*F*	*p*	Eta^2^
Cortisol awakening response (CAR)																		
AUC total	332.5	145.5	353.3	175.3	269.7	156.5	262.6	115.4	264.4	195.2	314.2	168.9	8.417	**0.001**	0.307	1.370	0.266	0.067
AUC basal	270.3	148.0	235.7	118.7	212.8	126.2	178.9	102.8	180.2	128.4	217.5	162.8	4.787	**0.014**	0.201	2.521	0.094	0.117
AUC net	62.2	81.5	117.6	105.9	56.0	78.5	83.8	76.7	84.2	106.1	96.8	82.6	2.181	0.127	0.103	0.800	0.457	0.040
sBDNF [ng/mL]	23.0	9.5	21.4	8.2	29.6	12.5	24.7	12.4	30.3	16.1	24.1	13.4	2.000	**0.014**	0.196	0.643	0.531	0.032
TNF-alpha [pg/mL]	12.0	8.4	12.5	8.3	13.3	9.1	10.9	8.3	12.7	9.1	11.9	8.0	0.023	0.977	0.001	1.258	0.295	0.061

Notes: *p*-values represent repeated measurements ANOVA, with a between subject factor “group” and a within subject factor “time”. Bold *p*-values: <0.05. Effect sizes are estimated by partial eta squared (η^2^). AUC = Area under the curve. sBDNF = Serum Brain Derived Neurotrophic Factor. TNF-alpha = Tumor Necrosis Factor alpha.

**Table 3 brainsci-11-00411-t003:** Sleep data.

	Baseline						Post intervention											
	Aerobic Exercise			Active Control Group			Aerobic Exercise			Active Control Group			Time			Time × Group		
	*n*	*M*	*SD*	*n*	*M*	*SD*	*n*	*M*	*SD*	*n*	*M*	*SD*	*F*	*p*	Eta^2^	*F*	*p*	Eta^2^
PSQI score	22	10.5	4.0	***19***	12.2	3.9	22	7.2	2.9	***19***	8.5	4.4	26.582	**<0.001**	0.412	0.111	0.741	0.003
Stage 1 [%]	19	9.6	6.0	17	7.5	4.8	13	7.7	5.6	14	4.9	2.4	3.998	0.057	0.138	0.049	0.826	0.002
Stage 2 [%]	19	66.2	9.9	17	66.1	7.0	13	67.5	10.2	14	67.9	8.3	0.177	0.677	0.007	0.014	0.096	0.001
Stage SWS [%]	19	4.6	5.6	17	5.7	7.0	13	6.2	6.3	14	7.2	7.2	2.078	0.162	0.077	0.807	0.378	0.031
Stage REM [%]	19	17.7	7.5	17	18.4	6.3	13	16.6	9.2	14	17.9	4.6	0.006	0.937	0.000	0.666	0.422	0.026
Sleep efficiency [%]	19	84.0	9.9	17	84.0	8.8	13	80.5	11.6	14	85.6	9.4	0.158	0.695	0.006	0.633	0.434	0.025
TST [min]	19	410.0	66.3	17	410.6	55.0	13	378.2	82.0	14	412.4	71.2	1.920	0.178	0.071	0.964	0.336	0.037
SOL [min]	19	20.4	14.8	17	22.9	15.5	13	49.2	47.0	14	28.0	39.3	3.416	0.076	0.120	1.870	0.184	0.070
WASO [min]	19	54.7	48.4	17	53.5	44.4	13	40.4	25.4	14	37.9	38.0	2.182	0.152	0.08	0.077	0.783	0.003
REM latency [min]	18	170.4	71.7	17	181.7	110.0	13	156.2	85.0	14	144.4	72.4	2.159	0.155	0.083	1.122	0.300	0.045
Awakenings	19	27.0	16.0	17	26.5	19.1	13	22.7	10.6	14	17.1	10.1	4.985	**0.035**	0.166	1.751	0.198	0.065

*p*-values represent repeated measurements ANCOVA, with a between subject factor “group” and a within subject factor “time”. Bold *p*-values: <0.05. Effect sizes are estimated by partial eta squared (η^2^). PSQI = Pittsburgh Sleep Quality Index. SWS = Slow wave sleep. TST = Total sleep time. SOL = Sleep onset latency. WASO = Wake after sleep onset. REM = Rapid eye movement.

**Table 4 brainsci-11-00411-t004:** Pearson’s Correlations.

		HDRS17 Post	PSQI Baseline	PSQI Post	REM Latency Post
dVO_2_max	***r***	−0.322	−0.375	−0.599	−0.437
	***p***	**0.049**	**0.024**	**<0.001**	**0.010**
TNF-alpha pre	***r***	−0.158	0.327	0.029	−0.076
	***p***	0.317	**0.039**	0.859	0.661

Notes: Pearson’s correlations. Bold *p*-values: <0.05. dVO2max = Change in aerobic capacity from baseline to post-intervention. TNF-alpha = Serum Tumor Necrosis Factor-alpha. HDRS17 = 17-item Hamilton Depression Rating Scale. PSQI = Pittsburgh Sleep Quality Index. REM = Rapid eye movement.

## Data Availability

The datasets used and analyzed during this study are available from the corresponding author on reasonable request.

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
