# Peer review of "Aerobic Exercise and Stretching as Add-On to Inpatient Treatment for Depression Have No Differential Effects on Stress-Axis Activity, Serum-BDNF, TNF-Alpha and Objective Sleep Measures"

_brainsci, 2021, doi:10.3390/brainsci11040411_

Round 1

Reviewer 1 Report

The submitted paper reports the results of a RCT comparing the effects of aerobic exercise (AE) versus an active control as add-on to standard treatment for depression on a few biological variables which might be relevant for the effects of antidepressant treatments (including AE), and which were considered as secondary outcomes in the design of the study.

The study is well conducted, the rationale for the study is clear and the results are presented in a clear and consistent fashion. Moreover, further information on this study is available in two previously published articles, one reporting detailed information about the protocol and the other reporting the results of the primary outcome of the study.

Importantly, the limitations of this study are clearly acknowledged in the Discussion.

I have no major concerns to raise about this article.

Below my observations.

  1. In my opinion, the previous article dealing with the primary outcome of this study should be cited in the Introduction (Imboden et al. 2020, ref. 31).

  2. Line 282: “we also found a statistical trend for time by group interaction of moderate effect size (p =0.094; Eta² = 0.117).” As far as I can understand, p does not denote effect size. I would use the expression “close to statistical significance”. Same for line 343 of Discussion.

  3. Line 394: maybe the clarity of this paragraph would benefit by a brief reference to the previous paper on the primary outcome of this study, since measures of depressive symptoms are not reported in this paper. [“Our findings suggest that improvement of maximum oxygen capacity (VO2max) is associated with reduced depressive symptoms.”]

Moreover:

  1. The greek characters α (as in line 208) and η (as in table 2) should be consistently used throughout the text (or, alternatively, alpha and eta).

  2. HDRS : Hamilton Depression Rating Scale (as in line 178): check abstract, line 19.

  3. line 111: ration should be ratio.

Author Response

Reviewer 1

The submitted paper reports the results of a RCT comparing the effects of aerobic exercise (AE) versus an active control as add-on to standard treatment for depression on a few biological variables which might be relevant for the effects of antidepressant treatments (including AE), and which were considered as secondary outcomes in the design of the study.

The study is well conducted, the rationale for the study is clear and the results are presented in a clear and consistent fashion. Moreover, further information on this study is available in two previously published articles, one reporting detailed information about the protocol and the other reporting the results of the primary outcome of the study.

Importantly, the limitations of this study are clearly acknowledged in the Discussion.

I have no major concerns to raise about this article.

Reply. Thank you for your appreciative and encouraging comments on the quality of our paper.

In my opinion, the previous article dealing with the primary outcome of this study should be cited in the introduction (Imboden et al. 2020, ref. 31).

Reply. Thank you. In line with your recommendation, we have included this reference in the introduction. It is now Reference 11.

Line 282: “we also found a statistical trend for time by group interaction of moderate effect size (p =0.094; Eta² = 0.117).” As far as I can understand, p does not denote effect size. I would use the expression “close to statistical significance”. Same for line 343 of Discussion.

Reply. Thank you for this comment. We fully agree with you. The effect-size refers to Eta2 not to the p-value. According to your suggestion, we changed the wording from “statistical trend” to “close to statistical significance (see lines 294-295 and 354 - 355).

Line 394: maybe the clarity of this paragraph would benefit by a brief reference to the previous paper on the primary outcome of this study, since measures of depressive symptoms are not reported in this paper. [“Our findings suggest that improvement of maximum oxygen capacity (VO2max) is associated with reduced depressive symptoms.”]

Reply. Thank you for this suggestion. We have added a sentence citing the previous publication and hope this enhances the clarity of the paragraph.

The Greek characters α (as in line 208) and η (as in table 2) should be consistently used throughout the text (or, alternatively, alpha and eta).

Reply. Thank you. Somehow the font “symbol” was corrupted during the uploading process (the document was written in Word for Mac OS). We have addressed this issue and now refer to TNF-alpha throughout the document without using the Greek symbol. We did the same for all other Greek symbols (Chi and Eta).

HDRS: Hamilton Depression Rating Scale (as in line 178): check abstract, line 19.

line 111: ration should be ratio.

Reply. Thank you. We have addressed these two issues. HDRS17 is now consistently spelled out “17-item Hamilton Depression Rating Scale”. “Ration” has been changed to “ratio”.

Reviewer 2 Report

In this manuscript, the authors compared the effect of AE and stretching (low-intensity activity) on outcome measures of the patients with depression undergoing treatment with anti-depressants. The authors did not find any significant differences between them although both have positive effects on the outcome measures. The study is very straight forward with a novel idea of investigating AE as add-on to anti-depressant treatment. I have only few minor comments/suggestions:

  • Why did authors choose to collect any data (2 weeks, for example) prior to 6 weeks?
  • Put values on x-axis without any decimal values whenever possible.
  • Write either TNF-a or TNF-alpha instead of TNF.

Author Response

Reviewer 2

In this manuscript, the authors compared the effect of AE and stretching (low-intensity activity) on outcome measures of the patients with depression undergoing treatment with anti-depressants. The authors did not find any significant differences between them although both have positive effects on the outcome measures. The study is very straight forward with a novel idea of investigating AE as add-on to anti-depressant treatment. I have only few minor comments/suggestions:

Reply. Thank you for your positive comment on our study and the novelty of its design.

Why did authors choose to collect any data (2 weeks, for example) prior to 6 weeks?

Reply. We measured several variables (CAR, sBDNF, TNF-alpha) after 2 weeks to detect early response / change of the variable, which allows a better understanding of the temporal dynamics of these variables. Based on your feedback, we have specified this in the methods section (lines 190-191).

Put values on x-axis without any decimal values whenever possible.

Reply. Thank you for this observation. We have adapted all figures accordingly.

Write either TNF-a or TNF-alpha instead of TNF.

 Reply. Thank you for this comment. We have adjusted this throughout the document and now consistently use the term “TNF-alpha”.

Reviewer 3 Report

This manuscript assesses the effect of aerobic exercise in patients with severe or moderate depression on the concentrations of BDNF, TNF, Cortisol Awakening Reaction, and objective sleep parameters measured with PSG.
Although the study is interestingly planned, it has many weaknesses.

-  There are no clearly presented objectives in the summary
- more information about neurotrophins and BDNF should be described, including the correlation between BDNF and exercise, depression, or sleep disorders (for example 10.2147/ndt.s5700; 10.3109/07853890.2015.1131327)
- introduction should be rewritten because it is a little unclear
- the abbreviations should be re-verified and explained
- the issue about BDNF, TNF, inflammation, and sleep should be better linked (for example 10.3389/fncel.2014.00430; 10.1111/nmo.13978)
- the aims are too long, unclear. It is unnecessary to cite the papers for the purpose section.
- the intensity of exercise was defined in a very subjective way. For example, an actigraph could be used to make this parameter more objective.
- the study group is very diverse. For example, patients with bipolar disorder and those treated with lithium are included. There are reports of the possible effects of lithium on BDNF concentration.
- how was the normal distribution calculated?
- PSG was not performed in all patients. Such a statistical analysis in which the number of participants is different at two-time points is not valuable.
- it is very unclear how many participants were eventually included in the analysis.
- the PSG is not crucial in this study. It can rather be used to eliminate diseases such as OSA or RLS, and not to diagnose insomnia, which is most often associated with depression. PSG evaluates a random night, so e.g. sleep duration may not represent the daily quality of sleep in these patients
- the results should be rewritten
- the TNF issue should be better described. It seems that these laboratory determinations do not bring much new value in their current form
- including the third group that did not exercise will be a good complement to this study

Author Response

Reviewer 3

This manuscript assesses the effect of aerobic exercise in patients with severe or moderate depression on the concentrations of BDNF, TNF, Cortisol Awakening Reaction, and objective sleep parameters measured with PSG.
Although the study is interestingly planned, it has many weaknesses.

Reply. Thank you for your overall positive feedback. We hope that we were able to address all your concerns by revising the paper in accordance with your recommendations and by discussing the weaknesses of its design appropriately.

There are no clearly presented objectives in the summary.

Reply. Thank you. Following your advice, we have added a sentence on the objectives of our study into the abstract (lines 16-18). In the main manuscript, the objectives are described in detail at the end of the introduction (lines 104-113).

More information about neurotrophins and BDNF should be described, including the correlation between BDNF and exercise, depression, or sleep disorders (for example 10.2147/ndt.s5700; 10.3109/07853890.2015.1131327)

Reply. Thank you for this suggestion. This is a good point. We have added a paragraph on the interplay between BDNF, stress, depression and insomnia in the introduction (see lines 59-63).

Introduction should be rewritten because it is a little unclear.

Reply. Thank you for your feedback. Without more concrete indications, it was difficult for us to revise the introduction. Based on your comment, we have carefully reread the introduction with a focus on clarity and hope that we were able to improve it, by incorporating your other suggestions concerning the introduction.

The abbreviations should be re-verified and explained.

Reply. Thank you for this observation. In the revised version of our manuscript, we have ensured that every abbreviation is included in the list of abbreviations.

The issue about BDNF, TNF, inflammation, and sleep should be better linked (for example 10.3389/fncel.2014.00430; 10.1111/nmo.13978).

Reply. Thank you for this suggestion. In line with your recommendation, we have added the topic of the interplay between inflammation, BDNF and depression in the introduction and cited the paper by Calabrese et al. (2014) that you mentioned in your comment (see lines 81-84).

The aims are too long, unclear. It is unnecessary to cite the papers for the purpose section.

Reply. Thank you for your comment. Per your request, we have shortened the purpose section in order to ensure that it now focuses more strongly on the aims. However, we believe that it is important to refer to previous research in this section if hypotheses are formulated. Since the interrelation of cardiorespiratory fitness and improved depression severity is a replication of earlier findings and we refer to earlier data, we believe that it is essential to cite this data in the section.

The intensity of exercise was defined in a very subjective way. For example, an actigraph could be used to make this parameter more objective.

Reply. Thank you. We accept this criticism and agree that an actigraph could have provided more objective information on the intensity of exercise in the control group. However, the program of the active control-group was highly standardized as we have pointed out in the methods section (see lines 161-163). Moreover, the intensity of exercise in the AE group was defined by objective parameters (heartrate, see lines 155-160). In line with your comment, we discussed this issue in the limitations section (see lines 467-469).

the study group is very diverse. For example, patients with bipolar disorder and those treated with lithium are included. There are reports of the possible effects of lithium on BDNF concentration.

Reply. Thank you. We agree that we have studied a heterogenous sample. The bipolar-issue has already been discussed thoroughly under limitations (see lines 453-458). We have also added a new paragraph in the revised document concerning the possible neurotrophic effects of lithium and hope that this addresses your concern adequately (see lines 455-460).

How was the normal distribution calculated?

Reply. Thank you for this comment. In fact, we evaluated normal distribution in SPSS using Q-Q-diagrams and by assessing skewness and kurtosis of all variables. We only found 2 demographic variables to seriously violate normality (“number of previous depressive episodes”, “number of previous maniac episodes”). Since these variables are not used as outcome-variables in this paper, we believe that this does not constitute a major issue. To further address this issue during the statistical analyses, we have logarithmized these two variables (Lg10). After this, skewness and kurtosis values were in an acceptable range (skewness <2, kurtosis <7). Therefore, we calculated the p-values in Table 1 with the logarithmized variables and pointed this out in the Notes.

PSG was not performed in all patients. Such a statistical analysis, in which the number of participants is different at two-time points is not valuable.

Reply. We agree that this decreases the power of the data analyses. Based on your comment, we have included a sentence in the limitations section, addressing this problem (lines 441-442). However, we believe that it is important to publish all data that were obtained during the data assessment.

It is very unclear how many participants were eventually included in the analysis.

Reply. Thank you for this comment. We believe that we have described the set for analysis (ITT sample n = 42) adequately in the results section (see lines 270-271) and in each table. We have also pointed out where we differed from the ITT-sample (namely for PSG-data). Table 3 clearly indicates the exact number of participations per group.

The PSG is not crucial in this study. It can rather be used to eliminate diseases such as OSA or RLS, and not to diagnose insomnia, which is most often associated with depression. PSG evaluates a random night, so e.g. sleep duration may not represent the daily quality of sleep in these patients.

Reply. Thank you for this comment. To the best of our knowledge, PSG data and changes in sleep architecture (such as REM-latency and SWS) are common biomarkers that have been studied previously in depression research. It is also common procedure to record only one night. Since we are studying inpatients, they are already sleeping in a changed environment for several nights at the time of the PSG, therefore chances to record impaired sleep quality due to a changed environment are very low. Based on your feedback, we have added two sentences addressing this under limitations (lines 435 - 439)

The results should be rewritten

Reply. Thank you for your feedback. Nevertheless, without more specific recommendations, we cannot know what we should change in the results section. In our opinion, we have appropriately described displayed the results of our study.

The TNF issue should be better described. It seems that these laboratory determinations do not bring much new value in their current form.

Reply. Thank you. We have changed the purpose section to better describe the reason why we have incorporated TNF-alpha as an outcome variable that might act as a predictor for antidepressant effects of AE as shown by Rethorst et al. (2013) (see lines 112-113). We have also included a sentence in the discussion section, stating that TNF-alpha might not be an adequate laboratory marker to detect anti-inflammatory effects of aerobic exercise (see lines 383-385). We hope that this and the changes in the introduction concerning TNF-alpha addresses your concern appropriately.

Including the third group that did not exercise will be a good complement to this study

Reply. We absolutely agree that such a group would further enhance the interpretation of the study results. However, we cannot change this fact because there was no TAU group in our study. We have addressed this issue at the end of the discussion (see lines 467-469).

Round 2

Reviewer 3 Report

The author addressed all of my comments. However, I have a few more comments:
- As TNF has also been studied, the issue of inflammation and BDNF should be better described. I recommend describing the role of BDNF in immune-related diseases, which are associated with a higher frequency of depression (10.1111/nmo.13978; 10.4061/2011/650685)
- Abbreviations should still be corrected. For example, "SSRI" is written for the first time on line 86 and this abbreviation is explained on line 175
- PSQI evaluates the last 4 weeks of sleep. So the study participants, when assessing their sleep at the second time point, did not really evaluate sleep after the intervention, but during it?
- PSG is not always a good method of assessing sleep. The hospital environment is not the only factor that influences its results, but also the equipment used to perform PSG. This is probably also one of the reasons why no differences were observed, e.g. in sleep time or in sleep architecture, between the two-time points. Sleep diaries would be more valuable in this study. This should be included in the discussion.
- line 325: Table 3? Not table 4?
- What were the assay ranges for the kits used to determine BDNF and TNF serum levels?

Author Response

Reviewer 3

Comments and Suggestions for Authors

The author addressed all of my comments. However, I have a few more comments:

Reply. Thank you for accepting our replies to your first review. We hope we were able to address your further comments in this second revision.

As TNF has also been studied, the issue of inflammation and BDNF should be better described. I recommend describing the role of BDNF in immune-related diseases, which are associated with a higher frequency of depression (10.1111/nmo.13978; 10.4061/2011/650685)

Reply. Thank you for this comment and the provided references. We have included a section on the role of BDNF in immune-related diseases such as Crohn’s disease in the introduction (Lines 84 – 88)

Abbreviations should still be corrected. For example, "SSRI" is written for the first time on line 86 and this abbreviation is explained on line 175

Reply. Thank you for this helpful observation. We thoroughly went through the manuscript and adapted all the abbreviations.

PSQI evaluates the last 4 weeks of sleep. So, the study participants, when assessing their sleep at the second time point, did not really evaluate sleep after the intervention, but during it?

Reply. Thank you for this comment. We agree that this may limit accuracy of PSQI at post-intervention. We have added a sentence about this limitation in the discussion (see lines 450 - 452)

PSG is not always a good method of assessing sleep. The hospital environment is not the only factor that influences its results, but also the equipment used to perform PSG. This is probably also one of the reasons why no differences were observed, e.g. in sleep time or in sleep architecture, between the two-time points. Sleep diaries would be more valuable in this study. This should be included in the discussion.

Reply. Thank you. We agree that there are limitations to PSG, and sleep diaries would have been a good addition to our sleep data. However, we used the same equipment in both study-centers and at each timepoint as described in the methods section (SomtéPSG). To better address this concern, we have added a sentence to the discussion section (lines 444 - 447)

line 325: Table 3? Not table 4?

Reply. Thank you for your thoroughness. This is absolutely correct. We have changed this to “Table 4”.

What were the assay ranges for the kits used to determine BDNF and TNF serum levels?

Reply. Thank you for this comment. We have added this information in the methods-section (lines 215 - 225)